# Estimating road traffic impacts of commute mode shifts

**Yue Hu**[1]*, **William Barbour**[1], **Kun Qian**[3], **Christian Claudel**[3], **Samitha Samaranayake**[2], **Daniel B. Work**[1]*

**1** Vanderbilt University and Institute for Software Integrated Systems, Nashville, TN, United States of America, **2** Cornell University, School of Civil and Environmental Engineering, Ithaca, NY, United States of America, **3** The University of Texas at Austin, Cockrell school of engineering, Austin, TX, United States of America

* yue.hu@vanderbilt.edu (YH); dan.work@vanderbilt.edu (DBW)

**Data Availability Statement:** The data underlying the results presented in the study are available from https://www.census.gov/programs-surveys/acs.

**Funding:** S.S and D.W: National Science Foundation under Grant Nos. CIS-2033580, https://

## Abstract

This work considers the sensitivity of commute travel times in US metro areas due to potential changes in commute patterns, for example caused by events such as pandemics. Permanent shifts away from transit and carpooling can add vehicles to congested road networks, increasing travel times. Growth in the number of workers who avoid commuting and work from home instead can offset travel time increases. To estimate these potential impacts, 6-9 years of American Community Survey commute data for 118 metropolitan statistical areas are investigated. For 74 of the metro areas, the average commute travel time is shown to be explainable using only the number of passenger vehicles used for commuting. A universal Bureau of Public Roads model characterizes the sensitivity of each metro area with respect to additional vehicles. The resulting models are then used to determine the change in average travel time for each metro area in scenarios when 25% or 50% of transit and carpool users switch to single occupancy vehicles. Under a 25% mode shift, areas such as San Francisco and New York that are already congested and have high transit ridership may experience round trip travel time increases of 12 minutes (New York) to 20 minutes (San Francisco), costing individual commuters $1065 and $1601 annually in lost time. The travel time increases and corresponding costs can be avoided with an increase in working from home. The main contribution of this work is to provide a model to quantify the potential increase in commute travel times under various behavior changes, that can aid policy making for more efficient commuting.

## Introduction

Transportation networks are critical infrastructure networks that are essential for moving goods and people efficiently [1]. Consequently, the need to understand road congestion in urban transportation networks at city scales has been a cornerstone of transportation science for more than 50 years [2–10]. Changes in travel demand can have dramatic impacts on the congestion levels observed on roadways. For example, travel restrictions and home-quarantine

www.nsf.gov/awardsearch/showAward?AWD_ID=
2033580&HistoricalAwards=false D.W and Y.H:
National Science Foundation, under Grant Nos.
CMMI-1727785, https://www.nsf.gov/
awardsearch/showAward?AWD_ID=1727785 The
funders had no role in study design, data collection
and analysis, decision to publish, or preparation of
the manuscript.

**Competing interests:** The authors have declared
that no competing interests exist.

orders designed to manage the spread of COVID-19 [11] can result in a sharp reduction in road traffic [12] as well as public transit ridership [13, 14].

To prepare for potential long term impacts of commute mode shifts on transportation networks, it is important to understand how commute patterns respond to events. For example, if commute patterns recover to pre-event levels, one can expect traffic to similarly resume. However, some events such as pandemics could result in shifts away from high density travel modes (e.g., public transit or carpooling) and into *single occupancy vehicles* (SOVs), altering the number of vehicles on the road network [15, 16]. Works [17–19] analyze the daily vehicle commuting patterns in different ways, and works [20–23] analyze the commute behavior change under COVID-19. Our work take one step forward and asks the important question: how will the shifts in commute patterns impact the road traffic?

To determine the sensitivity of road traffic to potential long term mode shifts, this article answers to what extent mode shifts away from transit and carpool towards single occupancy vehicles will change traffic in major *metropolitan statistical areas* (metro areas) in the US. Historical passenger vehicle average commute travel times as a function of the number of vehicles used for commuting from 2013 to 2018 are shown for 118 metro areas in Fig 1a. In each metro area, when the number of vehicles are normalized by the network capacity, and the travel times are normalized by the free flow travel time, the metro areas can be placed on a *universal BPR model* (Fig 1b). The position of each metro area on the universal curve explains the sensitivity of the travel time ratio to changes in the number of vehicles in the network (relative to the network capacity). The slope of the curve (Fig 1b) determines the sensitivity of each metro area to changes in the capacity-normalized number of passenger vehicles. If more passenger vehicles are added to the roadway, metro areas move up along the universal BPR curve. Such a shift is shown in Fig 1c for a setting where 25% of carpool and transit users shift to SOV in each metro area.

Our main finding is that metro areas including San Francisco, New York, Los Angeles, Boston, Chicago, Seattle, and San Jose have estimated travel time increases between 2–20 minutes additional round-trip commute travel time per person under a 25% mode shift from transit and car pool to SOV. This additional travel time due to congestion has an estimated cost per metro area of $2.5–24 million dollars per day, assuming the the hourly value of the time lost is equal to the median wage ($19.14/hr) from the Bureau of Labor Statistics [24]. We note that these potential increases can be avoided if 3–17% of all commuters work from home instead of commuting. Monitoring closely road usage, mode shifts, and work from home rates during

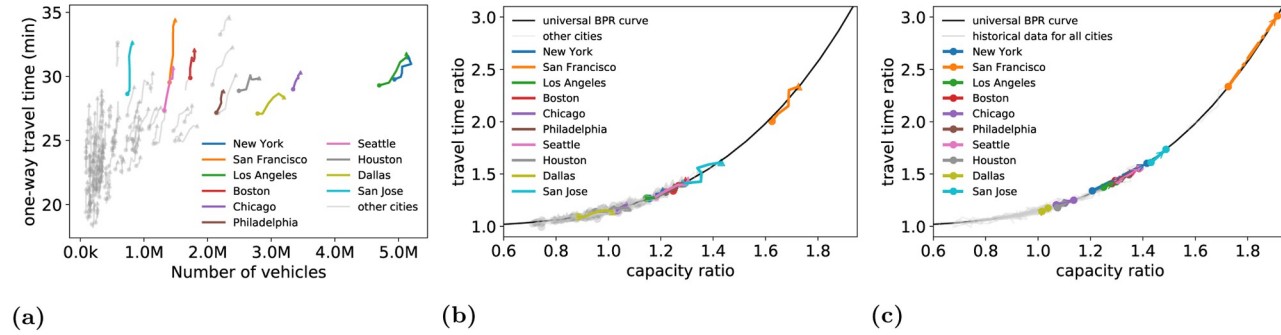

**Fig 1. (a)** One-way travel time vs. number of vehicles for 118 metro areas in the US from 2013–2018. **(b)** One way travel time ratio vs capacity ratio for 118 metro areas in the US. Metro areas appear on different portions of a universal BPR curve. For the two end points of each line, the dot denotes the 2013 conditions, while the triangle denotes the 2018 conditions. **(c)** Metro areas shift along the universal BPR curve under a 25% shift away from transit and carpool to SOVs. Grey lines correspond to all 118 metro areas in the dataset.

and and after mode shift triggering events will be important to detect and mitigate potential road traffic increases.

## Results

### Data: American Community Survey commute data

The ACS commute data contains commute statistics for each *metropolitan statistical areas* (MSA) defined by *The United States Office of Management and Budget* (OMB) [25]. Depending on when each MSA was introduced or when the boundaries of the MSA were last revised, up to nine years (2010—2018) of commute data records are available. The dataset contains the total number of people using each commute mode (e.g., 2-person carpool, 3-person carpool, single-occupancy vehicle, public transit, walk, and taxi/motorcycle/bike/other), as well as the average commute time for each mode. We select metro areas with at least six years of records under which the boundaries of the MSA were not altered in a way that resulted in a population change in excess of 5%. Under this restriction, sufficient data is available for 118 metro areas in total.

For each metro area each year, the total number of passenger vehicles used for commuting is computed by aggregating all vehicles used for two-person carpool, three-person carpool and single occupancy vehicles. The mean one-way commute travel time on the road is computed as the vehicle-weighted average of the travel times reported for single occupancy, two-person, and three-person carpool travel times.

Commuter data for taxis and ride hailing combined into a single category that also includes motorcycles and bikes. The influence of these modes on traffic is more challenging to model and vary depending on the vehicle type (e.g., bike compared to taxi) within the category. Because the combined category constitutes only a small portion of the total commuters (2% on average), the influence of these modes is ignored in the analysis that follows.

A plot of the 118 metro areas showing the trends between the number of vehicles and the one way travel time is shown in Fig 1. Several large MSAs are highlighted and labeled, while all remaining MSAs are shown in grey. Between 2013 (shown as a dot in Fig 1) and 2018 (shown as an arrow), the number of vehicles over time in each MSA tends to increase. One way travel times tend to increase as the number of vehicles grows within an MSA.

To determine if the number of passenger vehicles used for commuting in each metro area is a good predictor of the average commute travel time using BPR function, a correlation analysis between the one-way average commute travel time $\tau$ and the fourth order of traffic volume $N^4$ is conducted on all 118 metro areas. A total of 74 metro areas have a Pearson correlation coefficient of larger than 0.5 and two tailed significance $p$ value smaller than 0.1. This indicates that the BPR model (2) has prediction power for 74 metro areas. For the remaining 44 metro areas, traffic volume alone does not explain the historical variation of travel time. It is observed that all metro areas with low correlation between the number of vehicles and the travel time have a population of less than one million people, the largest being Columbus, OH (0.97 million people). In comparison, the metro areas with a correlation above 0.5 include most major metro areas in US, with an average population of 1.04 million. In the analysis that follows, we restrict data fitting and analysis to the 74 metro areas with correlation above 0.5.

### Approach: BPR model

We use the BPR model to describe the relationship between the number of passenger vehicles used for commuting and the corresponding average travel time. The BPR model [26, 27] is a classic model in the transportation engineering community that relates the volume of traffic on the road to the travel time to traverse it. The model captures the feature that when roads are

uncongested, adding vehicles to the road has negligible impact on travel times. However, once the roadway reaches its capacity, adding vehicles causes the travel time of all road users to increase. It is widely used in transportation management [28, 29], and network traffic simulation [30].

While the BPR model was originally designed to model travel times on a single road segment, recent studies have shown its applicability on urban scale transportation analysis [31, 32]. Like on individual road segments, the transition from free-flow to congested state characterized by a critical point is observed in [6, 33, 34]. Thus, the BPR model provides us with theoretical foundation of predicting metro area congestion based on traffic volume.

The BPR model reads

$$\tau = \tau_f \left( 1 + \alpha \left( \frac{N}{C} \right)^{\beta} \right), \tag{1}$$

where $\tau$ is the one-way average commute travel time and $N$ is the number of passenger vehicles on the roadway. The parameters $\tau_f$ and $C$ are the free flow travel time and the road network capacity respectively. Here, the capacity $C$ can be interpreted as the number of road users that can be accommodated in the city before average travel times quickly rise. Note that the average travel time $\tau$ is the average over the passenger vehicle commute times including at different times of the day. The shape parameters $\alpha$ and $\beta$ have a standard choice of $\alpha = 0.15$ and $\beta = 4.0$ [26, 27]. Under this choice, the model (1) reads

$$\tau = \tau_f + \theta(N^4), \tag{2}$$

with $\theta = 0.15\tau_f/C^4$. The form (2) shows that travel time $\tau$ and the fourth order of traffic volume $N^4$ have a linear relationship. Consequently, the model parameters $\tau_f$ and $\theta$ can be estimated from historic travel time data using linear methods. This in turn allows us to determine the free flow travel time $\tau_f$ and road capacity $C = \left( \frac{0.15\tau_f}{\theta} \right)^{\frac{1}{4}}$ for each city, deducing from (1).

We next introduce the universal BPR model. Let $\tilde{\tau}$ denote the travel time ratio computed as $\tilde{\tau} = \tau/\tau_f$ and $\tilde{N}$ as the capacity ratio $\tilde{N} = N/C$. The universal BPR model reads:

$$\tilde{\tau} = 1 + \alpha(\tilde{N})^{\beta}. \tag{3}$$

## Data analysis and BPR model parameter identification

Fitting the BPR model to the ACS data allows the estimation of the free flow travel time and the network capacity. For the 74 metro areas (full MSA names and shorthand names presented in S1 Table) with a correlation above 0.5, we use Bayesian linear regression [35, 36] to fit the BPR model (see Methods section). Bayesian linear regression provides the most likely prediction for travel time given traffic volume as well as a prediction distribution to measure the uncertainty of the prediction.

Table 1 and S2 Table contain the quantitative performance of the learned BPR model with respect to the *root mean square error* (RMSE) under *leave-one-out cross validation* (LOO-CV). For each of the 74 modeled metro areas, the RMSE is less than 1 min, and the average coefficient of determination ($R^2$) for all 74 modeled metro areas is 0.71, showing that the BPR model has good prediction power.

**Table 1. Summary of BPR models for 15 metro areas.** Years of data available for fitting the model and performance measures for the fitted model. See also S2 Table for a complete list.

|  | years of data | LOO RMSE (min) | $R^2$ score |
|---|---|---|---|
| New York | 6 | 0.48 | 0.73 |
| San Francisco | 9 | 0.73 | 0.92 |
| Los Angeles | 6 | 0.38 | 0.95 |
| Boston | 9 | 0.42 | 0.88 |
| Chicago | 9 | 0.30 | 0.78 |
| Philadelphia | 9 | 0.25 | 0.93 |
| Seattle | 9 | 0.32 | 0.97 |
| Houston | 9 | 0.52 | 0.86 |
| Dallas | 9 | 0.36 | 0.91 |
| San Jose | 9 | 0.93 | 0.87 |
| Atlanta | 9 | 0.38 | 0.92 |
| Miami | 9 | 0.37 | 0.94 |
| Portland | 9 | 0.36 | 0.95 |
| Riverside | 8 | 0.40 | 0.93 |
| Orlando | 9 | 0.60 | 0.85 |

The result of the Bayesian regression for all 74 modeled metro areas are shown in Fig 2, as well as S1–S4 Figs. The uncertainty of the model grows when the input data has larger noise and get more scattered, or when the traffic volume is further from available data.

## Marginal costs of additional road users

In this section we compute for each metro area the marginal cost of additional road users. The marginal cost is computed as

$$\frac{d\tau}{dN} = \beta \tau_f \alpha \left(\frac{N}{C}\right)^{\beta-1} \tag{4}$$

Similarly, the marginal cost of the universal BPR model reads:

$$\frac{d\tilde{\tau}}{d\tilde{N}} = \beta \alpha \tilde{N}^{\beta-1} \tag{5}$$

The marginal cost is the slope of the universal BPR model evaluated using the 2018 data for each metropolitan statistical area. It quantifies the sensitivity of the travel time ratio in the metro area with respect to changes in the capacity ratio. More concretely, it tells how quickly the travel time grows as a percentage of the free flow travel time, due to adding vehicles corresponding to a small fraction of the capacity. It is most meaningful to compare normalized impacts, because the capacities of the metro areas span multiple orders of magnitude. On the low end, metro areas like Rochester, NM have a capacity of 90k vehicles, while on the upper end, the New York metro area has a capacity of 5.16M vehicles. The addition of 9,000 vehicles to the roadway in Rochester consumes 1% of the network capacity, while the same 9,000 vehicles occupies 0.18% of the network in New York. Without normalization, small metro areas are more sensitive to each additional vehicle, due to the correspondingly larger portion of the capacity each vehicle consumes.

The marginal costs for each metro area appear in Table 2.

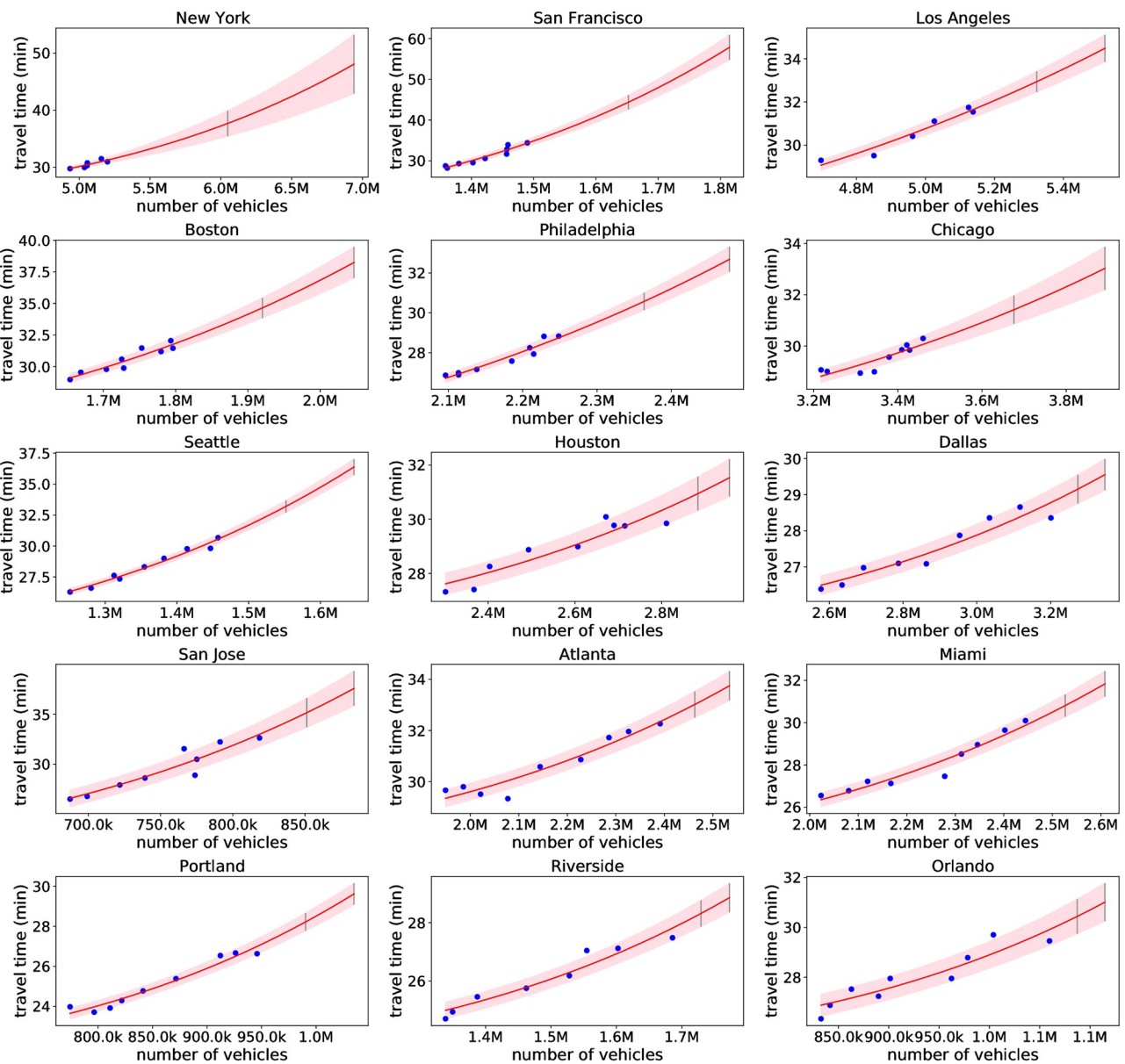

**Fig 2. Bayes fit for 15 metro areas.** The blue points are the observed data. The solid red line is the mean prediction, with shaded area covering ± one standard deviation of the prediction. Also shown are grey bars denoting the prediction intervals under a 25% (leftmost bar) and 50% (rightmost bar) transit and carpool mode shift to single occupancy vehicles. See also S1–S4 Figs for all 74 modeled metro areas.

## Congestion sensitivity to mode shifts away from transit and carpool

Using the calibrated BPR models for each MSA, this section considers the impact to traffic when the number of commuters stays the same, but a portion of transit and carpool users move into single occupancy vehicles.

To understand the sensitivity of the traffic conditions in each MSA to these mode shifts, we consider a scenario in which 25% of the carpool and transit users switch to single occupancy vehicles, and a second scenario in which 50% of the users switch. While the true mode shift

**Table 2. Marginal cost of additional users, quantifying the sensitivity of the travel time ratio with respect to changes in the capacity ratio.** Metro areas with the top 15 marginal costs are shown.

|  | passenger vehicles (M) | travel time (min) | capacity ratio | travel time ratio | marginal cost |
|---|---|---|---|---|---|
| San Francisco | 1.49 | 34.4 | 1.73 | 2.33 | 3.08 |
| San Jose | 0.82 | 32.6 | 1.43 | 1.61 | 1.76 |
| Oxnard | 0.35 | 24.6 | 1.43 | 1.61 | 1.75 |
| Seattle | 1.46 | 30.7 | 1.30 | 1.44 | 1.33 |
| Boston | 1.79 | 32.0 | 1.29 | 1.44 | 1.30 |
| Lancaster | 0.21 | 22.4 | 1.28 | 1.44 | 1.26 |
| Philadelphia | 2.25 | 28.8 | 1.28 | 1.41 | 1.26 |
| Charleston | 0.33 | 28.6 | 1.27 | 1.40 | 1.23 |
| Los Angeles | 5.13 | 31.7 | 1.25 | 1.37 | 1.17 |
| Stockton | 0.27 | 28.3 | 1.25 | 1.41 | 1.16 |
| Baton Rouge | 0.34 | 28.9 | 1.22 | 1.38 | 1.09 |
| Providence | 0.68 | 23.4 | 1.21 | 1.31 | 1.07 |
| New York | 5.16 | 31.5 | 1.21 | 1.34 | 1.06 |
| Baltimore | 1.14 | 30.2 | 1.20 | 1.32 | 1.03 |
| St. Louis | 1.21 | 26.1 | 1.19 | 1.31 | 1.00 |

that will be experienced in the future is unknown, the sensitivity approach allows to identify the metro areas that are the most sensitive to such mode shifts.

Table 3 provides a summary of the 2018 conditions in 15 metro areas (see also S3 Table for a complete list of all 74 modeled metro areas), including the total number of commuters, the number of passenger vehicles used for commuting (including SOVs and carpools), the number of transit riders, and the estimated one-way commute time by passenger vehicle. Table 3 also includes the prediction for each metro area when 25% of the transit and carpool commuters

**Table 3. Summary of 2018 transportation conditions for 15 metro areas, and changes when 25% of transit riders and carpools switch to single occupancy vehicles.** The range shows one standard deviation of the predictions. M denotes millions, and B denotes billions. See S3 Table for all 74 modeled metro areas.

|  | transportation conditions in 2018 | | | | prediction for 25% shift | | | |
|---|---|---|---|---|---|---|---|---|
|  | total commuters (M) | passenger vehicles (M) | transit riders (M) & (% of total commuters) | one-way travel time (min) | passenger vehicles (M) | one-way added time (min) | added $ cost per person per year | total added $ cost per year (B) |
| New York | 8.72 | 5.16 | 3.0(34.43%) | 31.00 | 6.05 | 6.7±2.2 | 1065.0±357.0 | 9.41±3.16 |
| San Francisco | 2.14 | 1.49 | 0.42(19.56%) | 34.30 | 1.65 | 10.0±1.8 | 1601.0±280.0 | 3.86±0.68 |
| Los Angeles | 5.92 | 5.13 | 0.31(5.24%) | 31.60 | 5.32 | 1.4±0.5 | 220.0±76.0 | 1.71±0.59 |
| Boston | 2.30 | 1.79 | 0.34(14.90%) | 31.70 | 1.92 | 3.0±0.8 | 470.0±128.0 | 1.32±0.36 |
| Chicago | 4.32 | 3.46 | 0.57(13.19%) | 30.10 | 3.68 | 1.4±0.6 | 216.0±88.0 | 1.16±0.47 |
| Philadelphia | 2.71 | 2.25 | 0.29(10.78%) | 28.80 | 2.36 | 1.8±0.4 | 291.0±69.0 | 1.0±0.24 |
| Seattle | 1.84 | 1.46 | 0.22(11.95%) | 30.60 | 1.55 | 2.6±0.5 | 422.0±77.0 | 0.96±0.17 |
| Houston | 3.10 | 2.81 | 0.06(2.09%) | 30.40 | 2.88 | 0.5±0.6 | 87.0±99.0 | 0.36±0.42 |
| Dallas | 3.49 | 3.20 | 0.05(1.42%) | 28.80 | 3.27 | 0.4±0.4 | 60.0±63.0 | 0.28±0.3 |
| San Jose | 0.95 | 0.82 | 0.04(4.26%) | 33.00 | 0.85 | 2.2±1.4 | 344.0±231.0 | 0.43±0.29 |
| Atlanta | 2.68 | 2.39 | 0.09(3.27%) | 32.40 | 2.46 | 0.7±0.5 | 107.0±81.0 | 0.38±0.29 |
| Miami | 2.77 | 2.44 | 0.09(3.38%) | 29.90 | 2.53 | 0.9±0.5 | 149.0±84.0 | 0.55±0.31 |
| Portland | 1.12 | 0.95 | 0.08(6.92%) | 27.00 | 0.99 | 1.2±0.4 | 195.0±71.0 | 0.28±0.1 |
| Riverside | 1.86 | 1.69 | 0.02(1.33%) | 27.80 | 1.73 | 0.5±0.4 | 80.0±72.0 | 0.2±0.18 |
| Orlando | 1.17 | 1.06 | 0.02(1.43%) | 29.90 | 1.09 | 0.5±0.7 | 84.0±111.0 | 0.13±0.18 |

switch to SOV. The total number of passenger vehicles under the switch is calculated by adding 25% of the 2018 transit riders and 25% of the carpools to the 2018 passenger vehicle count. The resulting number of passenger vehicles is then used as an input to the calibrated BPR model, and the one-way travel time forecast is produced. The difference between the 2018 baseline travel time and the new travel time under the switch is shown in Table 3. For example, in 2018, there were 8.72 million commuters in the New York metro area, of which 3.0 million (or 34.43%) were transit riders. The 5.16 million commuters taking a passenger vehicle (SOV or carpool) had an average commute travel time of 31.0 minutes. When 25% of the transit riders (750,000 commuters) and 25% of the carpool users (150,000) switch, a total of 900,000 additional passenger vehicles are used for commuting. This results in an increase of 6.7 minutes of commute time, up from 31.0 minutes to 37.7 minutes. The forecast standard deviation is 2.2 minutes.

The cost of the travel time increase is estimated per person and also across all passenger vehicle commuters within the MSA. Each cost estimate assumes the cost of an hour of time lost to commuting is the median hourly wage reported by the Bureau of Labor Statistics [24], following the practice of [37]. The most recent median hourly wage is $19.14/hr (May 2019). To compute the added cost per person per year, it is assumed each person has two commute trips each day (one from home to work, and one to return home from work), and works five days a week for 50 weeks each year. For the New York metro area, the 6.7±0.6 minutes of additional one-way commute travel times results in an increased cost per commuter of 1065±357 due to lost time alone.

To obtain the total added cost per day, the additional one-way passenger vehicle travel time due to mode shifts is doubled (assuming a round trip commute occurs each day), then multiplied by the value of time and the total number of passenger vehicle commuters. For the New York metro area, the 13.4 minutes of additional round trip commute delay experienced by 6.05 million passenger vehicle commuters results in a total daily cost of $25.78 million. A 25% increase is not equally likely in all cities. In places like NYC there are more barriers to switching away from transit due to costs (tolls, parking, etc.). This does not impact the model, but rather the amount of people that shift under the same epidemiological circumstances may differ from metro area to metro area.

The 15 metro areas shown in Table 3 are the metro areas with the largest total cost per day incurred due to a 25% mode shift. The New York metro area has the largest cost at $25.78 million due to the combination of a large travel time increase, and a large number of commuters experiencing the travel time increase. The San Francisco metro area has the highest travel time increase of 10.58 minutes of delay per commuter ($1601 annual cost per person), but a smaller total daily cost due to a smaller total number of passenger vehicles in the metro area. Seven of the 10 metro areas with the largest total cost per day have transit ridership levels in excess of 10%. Large (in number of commuters) metro areas with a large transit ridership (greater than 10%) have the most costly consequences of a mode shift.

The capacity, free flow travel time, *capacity ratio* (ratio of the number of passenger vehicles over the road capacity), and *travel time ratio* (ratio of the actual travel time over the free flow travel time) for the 15 most costly metro areas under a 25% mode shift are shown in Table 4 (See also S4 Table). By construction, all travel time ratios are greater than one, since the free flow travel time is defined as the travel time when the road is completely uncongested and empty. It shows how much longer a commute is due to the presence of traffic compared to an empty road (e.g., a travel time ratio of 1.15 means trips are 15% longer due to traffic) The capacity ratio can be less than one or greater than one, depending on if the network is loaded below the capacity, or above it. For each of the 15 most costly metro areas, the capacity ratios are all greater than one, ranging from 1.01 in Houston to 1.73 in San Francisco.

**Table 4. Estimated model parameters, capacity ratio, and travel time ratio for 15 metro areas.** (See also S4 Table for all 74 modeled metro areas).

|  | capacity (M) | free flow travel time (min) | capacity ratio (2018) | travel time ratio (2018) |
|---|---|---|---|---|
| New York | 4.27 | 23.5 | 1.21 | 1.34 |
| San Francisco | 0.86 | 14.7 | 1.73 | 2.33 |
| Los Angeles | 4.10 | 23.1 | 1.25 | 1.37 |
| Boston | 1.39 | 22.3 | 1.29 | 1.44 |
| Philadelphia | 1.75 | 20.5 | 1.28 | 1.41 |
| Chicago | 3.24 | 25.1 | 1.07 | 1.21 |
| Seattle | 1.12 | 21.3 | 1.30 | 1.44 |
| Houston | 2.62 | 25.3 | 1.07 | 1.18 |
| Dallas | 3.16 | 24.8 | 1.01 | 1.14 |
| San Jose | 0.57 | 20.3 | 1.43 | 1.61 |
| Atlanta | 2.23 | 27.0 | 1.07 | 1.20 |
| Miami | 2.08 | 23.2 | 1.17 | 1.29 |
| Portland | 0.80 | 20.9 | 1.18 | 1.27 |
| Riverside | 1.56 | 23.1 | 1.08 | 1.19 |
| Orlando | 0.99 | 25.0 | 1.07 | 1.18 |

The large estimated capacity ratio in San Francisco in 2018 is a result of the historical data for that metro area which is used to fit the BPR model. From 2013 to 2018, the number of passenger vehicles used for commuting in the San Francisco metro area rose by 6.2% (from 1.402 M vehicles to 1.490 M vehicles), while the corresponding commute travel time rose by 16.5% (from 29.53 min to 34.39 min). According to the BPR model, travel times grow more quickly for each additional vehicle added the further the network is loaded beyond the capacity (i.e., when it has a large capacity ratio). Comparatively, in the Los Angeles metro area, over the same period passenger vehicles used for commuting rose by 9.1% (from 4.697 M vehicles in 2013 to 5.125 M vehicles in 2018), while the corresponding travel times rose by 8.3% (from 29.29 to 31.74 min). The estimated capacity ratio for the Los Angeles metro area in 2018 is 1.25, which suggests that the road network is not loaded as far beyond the capacity compared to the San Francisco metro area. When compounded by the larger transit ridership in the San Francisco metro area compared to the Los Angeles metro area (both in absolute terms and as a percentage of total commuters), the road network San Francisco is more sensitive to a 25% mode shift away from transit.

To illustrate the range of impacts of a 25% mode shift away from carpool and transit to SOVs, Fig 1c shows the top 15 metro areas in terms of cost incurred under the 25% mode shift away from carpool and transit to SOVs. Each the number of passenger vehicles in each metro area are normalized by the network capacity to plot the capacity ratios. Travel times are similarly normalized and the resulting travel time ratios are shown. In grey, the historical data for all 74 modeled metros over all years of data are also shown, normalized by the estimated capacity and free flow travel time for each metro. The general trend from the historical data shows that travel time ratios grow more slowly in metro areas that are near or below capacity. After the number of passenger vehicles used for commuting in a metro area exceeds the capacity, travel time ratios grow more rapidly. The predictions under the 25% mode shift follow the same normalized curve defined by the historical data, with the change in the capacity ratio driven by how many commuters switch into SOVs. The growth in the travel time ratio is governed by how far beyond the capacity the network is (i.e., how far beyond 1 the capacity ratio is).

An analysis considering a 50% shift away from transit and carpool to SOVs is conducted similarly to the 25% mode shift. Due to the nonlinearity of the model a simple doubling of the number of transit and carpool users who switch into single occupancy vehicles leads to more than a doubling of minutes to the commute. For example, In the New York Metro area, the first 25% shift away from transit and carpool to SOV adds 6.2 minutes of delay to the average commute travel time. The next 25% shift adds an additional 9.3 minutes of delay to the average commute travel time. This is a consequence of the shape of the curve in Fig 1c), where the travel time (ratio) grows slowly at first, then more quickly as the capacity (ratio) continues to increase. Each passenger vehicle added to the road network has a higher marginal cost than the vehicle before it. The top 15 most costly metro areas are shown in Fig 3, with the percent increase compared to the 2018 baseline reported on each bar. The travel time increases under the 50% mode shift range from 4% for the Dallas metro area, to 68% for the San Francisco metro area.

## Offsetting factor: Increased rate of working from home

The increase in travel times due to mode shifts into SOVs can be offset by commuters who work from home instead of commuting to work. Table 5 (see also S5 Table) summarize the percentage of forecasted SOV users (baseline SOV users, and former transit and carpool users who would otherwise switch to SOV under the 25% or 50% switching rate) who must instead work from home to mitigate mode shift travel time increases. For example, in New York, if 17.22% of the potential passenger vehicle users instead work from home, the travel time will resume to 2018 levels even with a 25% shift away from transit and carpool. Considering a 25% shift, only the New York metro area and the San Francisco metro area require work from home rates in excess of 10% to offset potential travel time increases. At a more extreme 50% mode shift, the metro areas of Boston, Philadelphia, Chicago, and Seattle also require work from home rates in excess of 10% to offset the potential travel time increase.

## Discussion

Understanding the potential changes to traffic congestion if large scale mode shifts occur is important to maintain the efficient operation of road networks. This work provides a model to quantify the potential increase in commute travel times if a large portion of current transit and carpool commuters switch to SOVs. Using ACS data containing historical commute mode and road travel times for 118 MSAs, a BPR model is used to relate average commute times to traffic volume. Out of 118 MSAs with adequate data, 74 metros have travel times that are predictable using only the number of passenger vehicles on the road. These metro areas have a LOO-CV RMSE less than 1 minute, and an average r2 score of 0.7. The models can then be used to assess sensitivity of travel times to mode shift away from transit and carpool to SOV, as well as the work from home rate required to offset these increases.

There are several observations from the results. First, the BPR model captures that when the number of vehicles on the road increases, so does the travel time. But the increase in travel time for each added vehicle is not the same under different network congestion levels. Metro areas with networks already above capacity have more sensitive travel times to the addition of SOVs on the network. Metro areas such as San Francisco that are well above capacity and that also have a large number of transit users are most likely to see substantial travel time increases if mode shifts away from transit are realized. Even modest increases in delay per commuter per day manifest in millions of dollars of lost time for metro areas that have a large number of commuters that experience the delay.

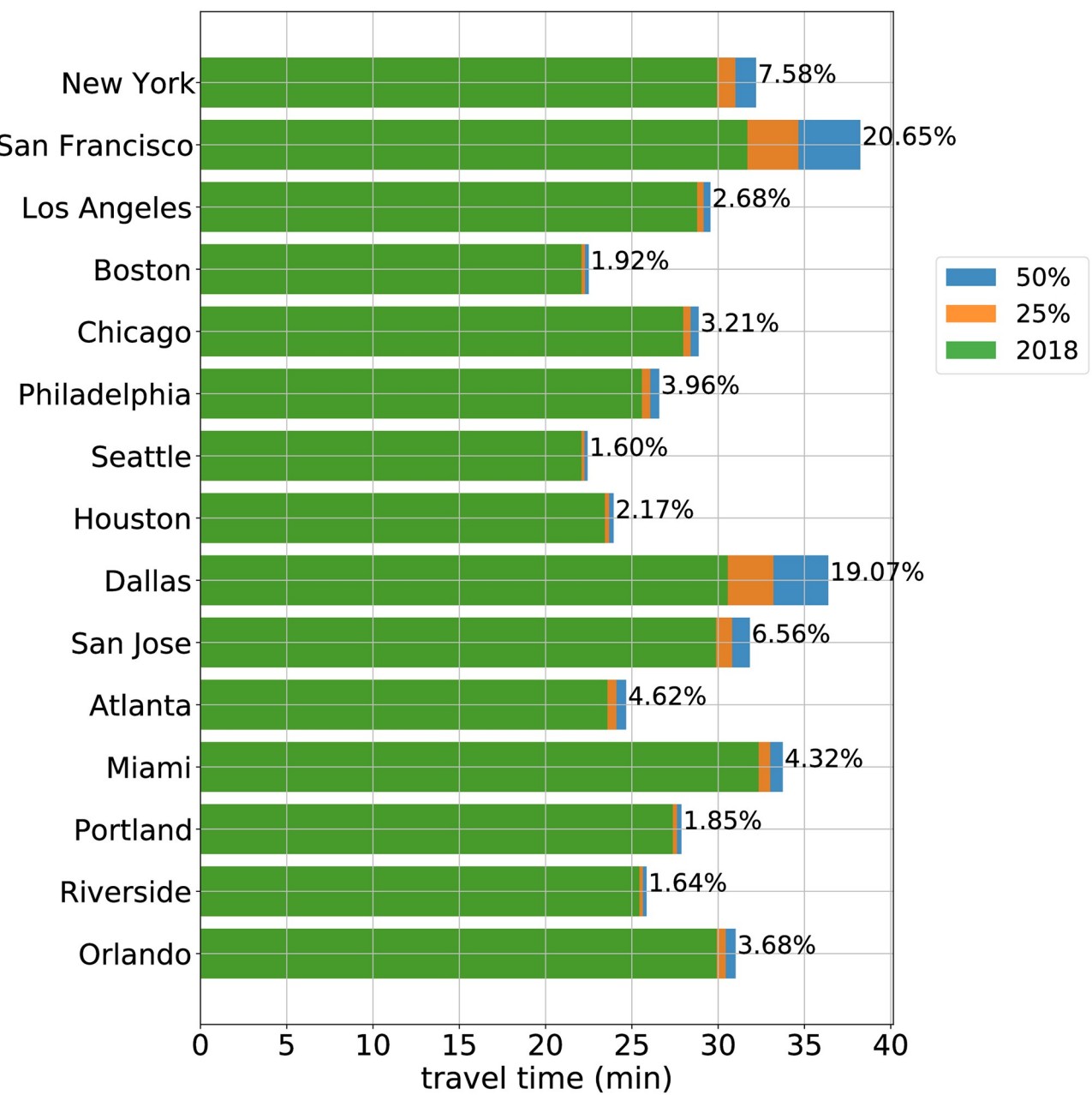

**Fig 3. Travel time increase predictions for top 15 metro areas in predicted cost.** The 2018 one-way travel time is shown in green. Also shown is the additional travel time under 25% (orange) and 50% shift (blue) from transit and carpool to SOV. The percentage increase of commute time from 2018 under a 50% shift appears to the right of each bar.

Second, it is important to note that travel time increases on road networks are avoidable. For example, in response to an event, transit ridership resumes in step with other modes, then traffic will similarly return to pre-event levels. Similarly, potential travel time increases can be avoided if work from home rates also increase. For the top 15 most costly metro areas under a 25% shift away from transit and carpool to SOV, 13 can avoid travel time increases when 2–7% of the passenger vehicle commuters work from home instead.

**Table 5. Percent of potential SOV commuters that are required to work-from-home to offset congestion increases due to 25% and 50% shifts away from transit and carpool into SOV.** See also S5 Table for all 74 modeled metro areas.

| | offsetting work from home rate | |
|---|---|---|
| | **25% transit and carpool shift** | **50% transit and carpool shift** |
| New York | 17.22% | 34.50% |
| San Francisco | 10.88% | 21.78% |
| Los Angeles | 3.76% | 7.61% |
| Boston | 7.27% | 14.35% |
| Philadelphia | 5.03% | 10.13% |
| Chicago | 6.24% | 12.49% |
| Seattle | 6.30% | 12.78% |
| Houston | 2.63% | 5.21% |
| Dallas | 2.32% | 4.60% |
| San Jose | 3.80% | 7.78% |
| Atlanta | 3.05% | 6.04% |
| Miami | 3.55% | 6.90% |
| Portland | 4.24% | 8.92% |
| Riverside | 2.32% | 4.90% |
| Orlando | 2.57% | 5.17% |

There are limitations to this work. The main limitation is that this work provides a sensitivity analysis of travel times given that a mode shift occurs, rather than a prediction specific mode shifts in response to an event. Knowing the true number of commuters who may switch modes at the level of each MSA is needed to design realistic scenarios to consider specific forecasts. The second limitation is that the analysis is based on the recent ACS data from 2018. The 2018 data is taken as the current baseline, and is not corrected for changes between 2018 and 2021 that influence both the baseline and the predictions. For metro areas that have increased the number of commuters without substantially increasing the road supply, the present results likely underestimate the travel times under the baseline and under mode shifts. The third limitation of the analysis is due to the spatial and temporal granularity at which it operates. The average commute time does not capture the variations in commute distances, routes, or the time of day of the commute. Depending on where, when, and how commutes occur in each metro area post-mode shift, some commuters will experience more direct impacts than others. There are other factors that influence commute travel times that we do not take into account, such as the road network configuration, the population density, and the distribution of trips over the duration of the day. While the number of vehicles alone can explain 74 of the metro areas, there are 44 metro areas that cannot make reliable travel time predictions using only the number of vehicles. These metro areas tend to be smaller in population. Fourth, the analysis considers the costs associated with lost time alone and is therefore a lower bound on the cost. More detailed accounting could also consider other costs of congestion such as extra fuel consumption or production of emissions.

In spite of the limitations described above, this work is the first quantified estimate that answers the question of how travel times in metro areas may be influenced by changes to commute patterns away from single occupancy vehicles. It provides estimates for 74 metro areas in the US, and provides insights into which areas are most sensitive to long term switches away from transit and carpool to SOV trips. The analysis can help mobility managers understand the factors that can change travel times when mode shift triggering events occur.

## Methods

### Bayesian linear regression

This section describes the Bayesian linear regression technique used to fit the BPR model to the available data. Bayesian linear regression provides both a most likely prediction of the travel time for a given number of vehicles, as well as a distribution of the uncertainty on the prediction. Bayesian linear regression is a widely used approach to linear regression, a comprehensive description can be found in [35, 36], and we document how we use Bayesian linear regression on the BPR model for completion.

Recall that in the BPR model (2), travel time $\tau$ is a linear function of the fourth order of number of vehicles $N^4$. We build a standard linear regression model for travel time and vehicle numbers as:

$$y = \mathbf{x}^T\mathbf{w} + \epsilon, \tag{6}$$

where $\mathbf{w} \in \mathbb{R}^2$ denotes the weight parameters, its two components $\mathbf{w}_1$ for slope and $\mathbf{w}_2$ for offset. $\mathbf{x} \in \mathbb{R}^2$ contains the input vehicle number $N^4$ and a constant; $y \in \mathbb{R}$ denotes the target output travel time $\tau$; and $\epsilon$ is a zero mean Gaussian distribution with precision (inverse variance) $\gamma$, $p(\epsilon) = \mathcal{N}(0, \gamma^{-1})$. For $n$ years of observations, We further denote $\mathbf{y} \in \mathbb{R}^n$ as all observed travel time data, and $\mathbf{X} \in \mathbb{R}^{2 \times n}$ as all vehicle number data. Following an i.i.d. sampling assumption, the distribution of observation output $\mathbf{y}$ given all observation input $\mathbf{X}$ can be written as:

$$p(\mathbf{y}|\mathbf{X}, \mathbf{w}) = \mathcal{N}(\mathbf{X}^T\mathbf{w}, \gamma^{-1}\mathbf{I}). \tag{7}$$

Next, in the fully Bayesian treatment of linear regression, we assume a zero mean isotropic Gaussian prior over the weight parameter $\mathbf{w}$, governed by a single precision parameter $\lambda$ [35]:

$$p(\mathbf{w}) = \mathcal{N}(\mathbf{w}|0, \lambda^{-1}\mathbf{I}). \tag{8}$$

Then, we learn the linear regression model given the observations, i.e., infer the posterior of $\mathbf{w}$ given $\mathbf{X}$ and $\mathbf{y}$ denoted as $p(\mathbf{w}|\mathbf{y},\mathbf{X})$, following Bayes' rule:

$$p(\mathbf{w}|\mathbf{y}, \mathbf{X}) = \frac{p(\mathbf{y}|\mathbf{X}, \mathbf{w})p(\mathbf{w})}{p(\mathbf{y}|\mathbf{X})}, \tag{9}$$

where the normalizing constant is the marginal likelihood given by [36]:

$$p(\mathbf{y}|\mathbf{X}) = \int p(\mathbf{y}|\mathbf{X}, \mathbf{w})p(\mathbf{w})d\mathbf{w}. \tag{10}$$

The posterior $p(\mathbf{w}|\mathbf{y},\mathbf{X})$ in (9) is proportional to the product of the likelihood $p(\mathbf{y}|\mathbf{X},\mathbf{w})$ and the prior $p(\mathbf{y}|\mathbf{X},\mathbf{w})$, and is also a Gaussian distribution. We use the standard procedure of completing the squares [35], and arrive at the posterior distribution:

$$p(\mathbf{w}|\mathbf{y}, \mathbf{X}) = \mathcal{N}(\mathbf{w}|\mathbf{m}, \mathbf{S}^{-1}), \tag{11}$$

where the mean value $\mathbf{m}$ and variance $\mathbf{S}^{-1}$ is given by:

$$\mathbf{m} = \gamma\mathbf{S}^{-1}\mathbf{X}\mathbf{y}, \tag{12}$$

$$\mathbf{S} = \gamma\mathbf{X}\mathbf{X}^T + \lambda\mathbf{I}. \tag{13}$$

Since the posterior distribution of $\mathbf{w}$ is Gaussian, the mode, or maximum a posterior (MAP) estimate $\mathbf{w}_{MAP}$, is the same as the mean value $\mathbf{m}$. From $\mathbf{w}_{MAP}$, we can calculate the BPR

parameters according to Equation (2). Road network capacity $C = \left(\frac{0.15\mathbf{w}_{MAP1}}{\mathbf{w}_{MAP1}}\right)^{\frac{1}{4}}$, and free flow travel time $\tau_f = \mathbf{w}_{MAP2}$, where slope parameter $\mathbf{w}_{MAP1}$ and offset parameter $\mathbf{w}_{MAP2}$ are the two components of $\mathbf{w}_{MAP}$. The result of BPR parameters are tabulated in Table 4 and S4 Table.

After learning the linear model parameters, we can predict the travel time for queried vehicle number $N_*$, such as when 25% of carpool and transit commuters shift to SOV as reported in Table 3 and S3 Table. Specifically, we construct a new input $\mathbf{x}_* \in \mathbb{R}^2$ containing $N_*^4$ and a constant. Given the posterior distribution of $\mathbf{w}$ and new input $\mathbf{x}_*$, we calculate the distribution for output value $y$ by averaging over all possible $\mathbf{w}$ values, weighted by their posterior probability [36]:

$$p(y|\mathbf{x}_*, \mathbf{X}, \mathbf{y}) \quad = \int p(y|\mathbf{x}_*, \mathbf{w})p(\mathbf{w}|\mathbf{y}, \mathbf{X})d\mathbf{w} \tag{14}$$

$$= \mathcal{N}\left(\mathbf{x}^T\mathbf{m}, \frac{1}{\gamma} + \mathbf{x}_*^T\mathbf{S}^{-1}\mathbf{x}_*\right). \tag{15}$$

$$= \mathcal{N}\left(\gamma\mathbf{x}_*^T\mathbf{S}^{-1}\mathbf{X}\mathbf{y}, \frac{1}{\gamma} + \mathbf{x}_*^T\mathbf{S}^{-1}\mathbf{x}_*\right). \tag{16}$$

The predictive distribution is a Gaussian distribution, where the variance comes from two sources. The first term $\frac{1}{\gamma}$ represents the noise of the data. The second term $\mathbf{x}_*^T\mathbf{S}^{-1}\mathbf{x}_*$ represents the uncertainty over the estimation of $\mathbf{w}$. As the magnitude $\mathbf{x}_*$ increases, so does the predictive uncertainty.

As a final note, the hyper-parameters $\gamma$, $\lambda$ are usually priors that are specified before seeing the data, which can be distributions themselves. But they can also be set to specific values by maximizing the marginal likelihood function integrated over the weight parameters $\mathbf{w}$ [35],

$$p(\mathbf{y}|\gamma, \lambda) = \int p(\mathbf{y}|\mathbf{w}, \gamma)p(\mathbf{w}|\lambda)d\mathbf{w}. \tag{17}$$

This framework is sometimes called empirical Bayes [38], and is adopted in our approach.

In this work, the computation of Bayesian linear regression is carried out using the python scikit-learn package [39]. For the regression of each city, the input data (the fourth order of number of vehicles $N^4$) is scaled to 0–1 interval before feeding in the linear regression model.

## Supporting information

**S1 Table. Common shorthand Metropolitan Statistical Area (MSA) name and the corresponding complete US Census Bureau (USCB) official metropolitan statistical area name.** (PDF)

**S2 Table. Summary of BPR models for all 74 analysed metro areas.** Years of data available for fitting the model and performance measures for the fitted model. (PDF)

**S3 Table. Summary of city transportation status in 2018 and prediction if one in four commuters switch from transit or car share mode to SOV mode.** All 74 analysed cites are shown, ranked by total cost per day. The range shows one standard deviation of the predictions. M denotes millions. B denotes billions. The major city is listed here in representation of the metro area. (PDF)

**S4 Table. Summary of inferred BPR model parameters for all 74 modeled metro areas.**
(PDF)

**S5 Table. Percent of adaptation to work-from-home needed to offset the influence of 25% and 50% transit shift to SOV, out of all potential SOV commuters.** Result for all 74 analysed cites are shown.
(PDF)

**S1 Fig. Bayes fit for 74 metro areas.** The blue points are the observed data. The solid red line is the mean prediction, with shaded area covering ± one standard deviation of the prediction. Also shown are grey bars denoting the prediction intervals under a 25% (leftmost bar) and 50% (rightmost bar) transit and carpool mode shift to single occupancy vehicles. (Part 1, continued in S2 Fig).
(TIFF)

**S2 Fig. Bayes fit for 74 metro areas.** The blue points are the observed data. The solid red line is the mean prediction, with shaded area covering ± one standard deviation of the prediction. Also shown are grey bars denoting the prediction intervals under a 25% (leftmost bar) and 50% (rightmost bar) transit and carpool mode shift to single occupancy vehicles. (Part 2, continued in S3 Fig).
(TIFF)

**S3 Fig. Bayes fit for 74 metro areas.** The blue points are the observed data. The solid red line is the mean prediction, with shaded area covering ± one standard deviation of the prediction. Also shown are grey bars denoting the prediction intervals under a 25% (leftmost bar) and 50% (rightmost bar) transit and carpool mode shift to single occupancy vehicles. (Part 3, continued in S4 Fig).
(TIFF)

**S4 Fig. Bayes fit for 74 metro areas.** The blue points are the observed data. The solid red line is the mean prediction, with shaded area covering ± one standard deviation of the prediction. Also shown are grey bars denoting the prediction intervals under a 25% (leftmost bar) and 50% (rightmost bar) transit and carpool mode shift to single occupancy vehicles. (Part 4).
(TIFF)

## Author Contributions

**Conceptualization:** William Barbour, Samitha Samaranayake, Daniel B. Work.

**Data curation:** William Barbour.

**Funding acquisition:** Samitha Samaranayake, Daniel B. Work.

**Investigation:** Yue Hu, Daniel B. Work.

**Methodology:** Yue Hu, Kun Qian, Samitha Samaranayake, Daniel B. Work.

**Project administration:** Samitha Samaranayake, Daniel B. Work.

**Supervision:** Christian Claudel, Samitha Samaranayake, Daniel B. Work.

**Visualization:** Yue Hu, William Barbour.

**Writing – original draft:** Yue Hu, Daniel B. Work.

**Writing – review & editing:** William Barbour, Kun Qian, Christian Claudel, Samitha Samaranayake.

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
