## [Decision Letter · Decision Letter 0]

12 Jun 2022

PONE-D-21-28489Estimating road traffic impacts of commute mode shiftsPLOS ONE

Dear Dr. Hu,

Thank you for submitting your manuscript to PLOS ONE. After careful consideration, we feel that it has merit but does not fully meet PLOS ONE’s publication criteria as it currently stands. Therefore, we invite you to submit a revised version of the manuscript that addresses the points raised during the review process. Both reviewers feel that this paper is of great significance, and please further revise the manuscript carefully according to the reviewers' comments.

We look forward to receiving your revised manuscript.

Kind regards,

Sheng Jin

Academic Editor

PLOS ONE

Journal Requirements:

Reviewers' comments:

Reviewer's Responses to Questions

**Comments to the Author**

1. Is the manuscript technically sound, and do the data support the conclusions?

Reviewer #1: Yes

Reviewer #2: Yes

2. Has the statistical analysis been performed appropriately and rigorously? 

Reviewer #1: Yes

Reviewer #2: Yes

3. Have the authors made all data underlying the findings in their manuscript fully available?

Reviewer #1: Yes

Reviewer #2: Yes

4. Is the manuscript presented in an intelligible fashion and written in standard English?

Reviewer #1: Yes

Reviewer #2: Yes

5. Review Comments to the Author

Reviewer #1: Understanding the potential changes to traffic congestion if large scale mode shifts occur is important to maintain the efficient operation of road networks. In general, the paper is well written and structured. However, the reviewer have some questions, and I hope the reviewer can address them or provide a reasonable explanation. My major comments are below:

1. The authors summarized the conclusion in the abstract, but what about the contribution in your study.

2. “We use the BPR model to describe the relationship between the number of passenger vehicles used for commuting and the corresponding average travel time”. It may be incorrect. I think all of vehicles, not just for commuters should be considered here.

3. Some related works should be added and discussed. For example, Understanding vehicles commuting pattern based on license plate recognition data.

4. All of figures are ambiguous.

Reviewer #2: The author has done a great job and there are 2 minor comments that I would like the author to revise.

First, the structure of the paper is similar to that of a comprehensive journal like nature communication, where the result is placed directly after the introduction. I have read other papers in plos one and most of them are not in this structure, so i suggest to adjust it.

Secondly, the latest reference is from 2020, which is relatively old, and there are fewer papers from transportation journals. I understand that the author's background is in computer science and has experience publishing in a top journal like TKDD. However, the topic of this paper is closely related to intelligent transportation systems (ITS), and the reviewers are in the field of ITS, so I suggest the authors cite at least three recent papers from journals in the field of ITS, e.g., Communications in Transportation Research and Journal of Intelligent and Connected Vehicles.

6. PLOS authors have the option to publish the peer review history of their article (what does this mean?). If published, this will include your full peer review and any attached files.

Reviewer #1: No

Reviewer #2: No

---

## [Author Response · Author response to Decision Letter 0]

8 Aug 2022

We have uploaded response to reviewers in a separate PDF. Also pasted the response below:

To Editor:

We thank the editor for the opportunity to revise the manuscript in light of the questions and comments from the reviewers. The reviewers noted that our paper address an important problem of understanding the potential changes to traffic congestion if large scale mode shifts occur (reviewer 1), and that the paper is well written (reviewer 1 and 2). Both reviewers made suggestion on adding more to the literature review, which we did accordingly. The reviewers also raised questions on the modeling, figure style and paper structure, which we also checked and explained respectively.

To Reviewer 1:

We thank the reviewer for the positive feedback, and address the major comments below.

1.We have added the contribution statement in the abstract. 

2.The reviewer is correct that we did not include all vehicles in the study. However, this is intentional. In this paper we regressed the commute time on the number of commuting vehicles. In this way, we provide a simple and elegant approach to estimate commute time based on information that we know, i.e., the number of passenger vehicles. The result shows that for most of the major metro areas in US, there is a strong correlation between commute travel time and number of passenger vehicles - details are reported in Section ``Results'', subsection ``Data: American Community Survey commute data''.

Moreover, by using a Bayesian linear regression model, the uncertainty of the prediction, e.g., brought by not including all vehicles besides commuting cars, is measured. We finally note that the total number of vehicles is unknown in the dataset, so it is not possible to regress based on this quantity.

3.We thank the reviewer for pointing out the related works. They are added.

4.We double checked that the figures are properly referenced, and that the units and captions are complete. We would be happy to further modify the figures if we have omitted critical information. 

To Reviewer 2:

1.We thank the reviewer for pointing out the potential manuscript organization concern. We note that according to the PLOS One guideline (https://journals.plos.org/plosone/s/submission-guidelines), our current organization of placing results directly after introduction is in line with the journal standards. Specifically, the guideline states that after beginning section (including abstract and introduction), in the middle section, the following elements can be renamed as needed and presented in any order: Materials and Methods, Results, Discussion, Conclusions (optional). There are also works in PLOS one that place results directly after introduction, like [1,2]. Therefore, we choose the current order for best readability.

2.We agree with the reviewer that the reference is relatively old. We have added new publications on related topic in 2021 and 2022. Works in the ITS field, e.g. from Communications in Transportation Research and IEEE Transactions on Intelligent Transportation Systems are included. 

[1]Mehmet Yildirimoglu and Osman Kahraman. Searching for empirical evidence on traffic equilibrium. PloS one, 13(5):e0196997, 2018

[2]Rohan L Aras, Nicholas T Ouellette, and Rishee K Jain. Automated identification of urban substructure for comparative analysis. Plos one, 16(1):e0245067, 2021

---

## [Decision Letter · Decision Letter 1]

14 Sep 2022

PONE-D-21-28489R1Estimating road traffic impacts of commute mode shiftsPLOS ONE

Dear Dr. Hu,

Thank you for submitting your manuscript to PLOS ONE. After careful consideration, we feel that it has merit but does not fully meet PLOS ONE’s publication criteria as it currently stands. Therefore, we invite you to submit a revised version of the manuscript that addresses the points raised during the review process.

Please focus on the comments of the second reviewer for detailed revisions.

We look forward to receiving your revised manuscript.

Kind regards,

Sheng Jin

Academic Editor

PLOS ONE

Reviewers' comments:

Reviewer's Responses to Questions

**Comments to the Author**

1. If the authors have adequately addressed your comments raised in a previous round of review and you feel that this manuscript is now acceptable for publication, you may indicate that here to bypass the “Comments to the Author” section, enter your conflict of interest statement in the “Confidential to Editor” section, and submit your "Accept" recommendation.

Reviewer #1: All comments have been addressed

Reviewer #3: (No Response)

2. Is the manuscript technically sound, and do the data support the conclusions?

Reviewer #1: Yes

Reviewer #3: Partly

3. Has the statistical analysis been performed appropriately and rigorously? 

Reviewer #1: Yes

Reviewer #3: Yes

4. Have the authors made all data underlying the findings in their manuscript fully available?

Reviewer #1: Yes

Reviewer #3: Yes

5. Is the manuscript presented in an intelligible fashion and written in standard English?

Reviewer #1: Yes

Reviewer #3: Yes

6. Review Comments to the Author

Reviewer #1: The authors have addressed my comments well. Now, I recommend to accept the paper with the current version.

Reviewer #3: This paper proposes a method to estimate the impact of commute mode shifts on commute travel time, which is an interesting and important topic. I mainly have the following comments:

Major suggestions:

1. I would like to know why the authors did the research only in MSA regions instead of modeling in all regions.

2. I am not sure whether the parameter determination logic of the BPR model is correct. I think the free flow travel time and road network capacity should be determined through data or prior knowledge, and then fitted α and β of BPR model, rather than determine the α and β first, then get free flow travel time and road network capacity by fitting. And the determination of α and β requires more basis.

3. The data used to calibrate the free flow travel time and road network capacity of the city span about 10 years. In 10 years, the free flow travel time and road network capacity of the same city may change greatly, which needs to be taken into consideration.

4. It is suggested to add a section of literature review, and give the methods used in similar researches and the advantages of this study.

Minor suggestions:

1. The “universal PBR model” in line 26 in Introduction should be changed to “universal BPR model”.

2. What does the “MSA” in line 49 in Results mean? No explanation is given. Is it the abbreviation of metropolitan statistical areas. The authors should check the full manuscript and explain any abbreviations and symbols before they first appear.

3. The explanations of τ and N4 in line 76 in Results are not given.

7. PLOS authors have the option to publish the peer review history of their article (what does this mean?). If published, this will include your full peer review and any attached files.

Reviewer #1: No

Reviewer #3: No

---

## [Author Response · Author response to Decision Letter 1]

18 Oct 2022

We have attached the respond to reviewers as a separate file, also pasting it here. 

Reviewer 3:

1. We choose to conduct the research on MSA regions, because the MSA regions have the most complete historical data available, provided by the U.S Census Bureau. Moreover, MSA covers regions with high population density, which are of most interest for our study on commuter pattern shifts.

2. In the BPR model, the choice of $\\alpha = 0.15$ and $\\beta = 0.4$ is standard, and wide studies and applications have seen its applicability[1-4]. In the mean time, the free flow travel time and the road network capacity across a whole region is hard to directly measure. Thus, we choose to fit the BPR model with fixed $\\alpha = 0.15$ and $\\beta = 0.4$, and derive the the free flow travel time and the road network capacity. We revised the manuscript to make this point more clear.

3. The author is correct that there might be changes in free flow travel time and road network capacity for a city. Yet when we fit the historical data of 118 metro areas to the BPR model, we found that 74 of the metro areas have a Pearson correlation coefficient of larger than 0.5 and two tailed significance $p$ value smaller than 0.1. This means that the influence of road network capacity shift is negligible for these 74 areas, and we restrict data fitting and analysis to the 74 metro areas in our work. 

4. Although we do not have a separate chapter on literature review, we have included the related works in the Introduction section. Specifically, works[5-7] analyze the daily vehicle commuting patterns in different ways, and works[8-11] analyze the commute behavior change under COVID-19. And our work take one step forward and asks the important question: how will the shifts in commute patterns impact the road traffic? There are no previous works to answer this question, and our approach is unique. 

Minor issues:

We thank the reviewer for pointing out the typo and lacking in definitions. We have fixed the manuscript accordingly.

[1] Gary A Davis and Hui Xiong. Access to destinations: travel time estimation on arterials. 2007.

[2] Wai Wong and SC Wong. Network topological effects on the macroscopic bureau of public roads function.

Transportmetrica A: Transport Science, 12(3):272–296, 2016.

[3] Rafal� Kucharski and Arkadiusz Drabicki. Estimating macroscopic volume delay functions with the traffic density derived from measured speeds and flows. Journal of Advanced Transportation, 2017, 2017.

[4] Hui-fang Tan, Yang Yang, and Ling-rui Zhang. Improved bpr function to counter road impedance through od matrix estimation of freight transportation. Journal of Highway and Transportation Research and Denelopment, 11(2):97–102, 2017.

[5] Wenbin Yao, Maolei Zhang, Sheng Jin, and Dongfang Ma. Understanding vehicles commuting pat- tern based on license plate recognition data. Transportation Research Part C: Emerging Technologies, 128:103142, 2021.

[6] Xiaolei Ma, Congcong Liu, Huimin Wen, Yunpeng Wang, and Yao-Jan Wu. Understanding commuting patterns using transit smart card data. Journal of Transport Geography, 58:135–145, 2017.

[7] Kevin S Kung, Kael Greco, Stanislav Sobolevsky, and Carlo Ratti. Exploring universal patterns in human home-work commuting from mobile phone data. PloS one, 9(6):e96180, 2014.

[8] Ziheng Huang, Dujuan Wang, Yunqiang Yin, and Xiang Li. A spatiotemporal bidirectional attention- based ride-hailing demand prediction model: A case study in beijing during covid-19. IEEE Transactions on Intelligent Transportation Systems, 2021

[9] Wenbin Yao, Jinqiang Yu, Ying Yang, Nuo Chen, Sheng Jin, Youwei Hu, and Congcong Bai. Under- standing travel behavior adjustment under covid-19. Communications in Transportation Research, page 100068, 2022.

[10] Shaila Jamal, Sadia Chowdhury, and K Bruce Newbold. Transport preferences and dilemmas in the post-lockdown (covid-19) period: Findings from a qualitative study of young commuters in dhaka, bangladesh. Case studies on transport policy, 10(1):406–416, 2022.

[11] Samuele Marinello, Francesco Lolli, and Rita Gamberini. The impact of the covid-19 emergency on local vehicular traffic and its consequences for the environment: The case of the city of reggio emilia (italy). Sustainability, 13(1):118, 2020.

---

## [Decision Letter · Decision Letter 2]

14 Dec 2022

Estimating road traffic impacts of commute mode shifts

PONE-D-21-28489R2

Dear Dr. Hu,

We’re pleased to inform you that your manuscript has been judged scientifically suitable for publication and will be formally accepted for publication once it meets all outstanding technical requirements.

Kind regards,

Sheng Jin

Academic Editor

PLOS ONE
---

## [Editor Report · Acceptance letter]

21 Dec 2022

PONE-D-21-28489R2 

Estimating road traffic impacts of commute mode shifts 

Dear Dr. Hu:

I'm pleased to inform you that your manuscript has been deemed suitable for publication in PLOS ONE. Congratulations! Your manuscript is now with our production department. 

Kind regards, 

on behalf of

Dr. Sheng Jin 

Academic Editor

PLOS ONE